# Conventional and Advanced Magnetic Resonance Imaging Assessment of Non-Enhancing Peritumoral Area in Brain Tumor

**DOI:** 10.3390/cancers15112992

**Published:** 2023-05-30

**Authors:** Elisa Scola, Guido Del Vecchio, Giorgio Busto, Andrea Bianchi, Ilaria Desideri, Davide Gadda, Sara Mancini, Edoardo Carlesi, Marco Moretti, Isacco Desideri, Giovanni Muscas, Alessandro Della Puppa, Enrico Fainardi

**Affiliations:** 1Neuroradiology Unit, Department of Radiology, Careggi University Hospital, 50134 Florence, Italy; scolae@aou-careggi.toscana.it (E.S.);; 2Radiodiagnostic Unit N. 2, Department of Experimental and Clinical Biomedical Sciences, University of Florence, 50121 Florence, Italy; 3Radiation Oncology, Oncology Department, Careggi University Hospital, University of Florence, 50121 Florence, Italy; 4Neurosurgery Unit, Department of Neuroscience, Psychology, Pharmacology and Child Health, Careggi University Hospital, University of Florence, 50121 Florence, Italy; 5Neuroradiology Unit, Department of Experimental and Clinical Biomedical Sciences “Mario Serio”, University of Florence, 50121 Florence, Italy

**Keywords:** MRI, advanced techniques, brain tumors, peritumoral edema, neuroradiology

## Abstract

**Simple Summary:**

The non-enhancing peritumoral area (NEPA) is defined as the hyperintense region in T2-weighted and fluid-attenuated inversion recovery (FLAIR) images surrounding a brain tumor. Analyses of the NEPA provide additional information to the magnetic resonance imaging (MRI) evaluation of the enhancing part of brain tumors. NEPA analyses of central nervous system malignancies with conventional and advanced MRI techniques give interesting highlights in the differential diagnosis of solid contrast-enhanced brain tumors regarding the prognostic stratification and treatment response. Considering the rapid technological changes in advanced MRI imaging, we assume that an update on this topic might be of interest.

**Abstract:**

The non-enhancing peritumoral area (NEPA) is defined as the hyperintense region in T2-weighted and fluid-attenuated inversion recovery (FLAIR) images surrounding a brain tumor. The NEPA corresponds to different pathological processes, including vasogenic edema and infiltrative edema. The analysis of the NEPA with conventional and advanced magnetic resonance imaging (MRI) was proposed in the differential diagnosis of solid brain tumors, showing higher accuracy than MRI evaluation of the enhancing part of the tumor. In particular, MRI assessment of the NEPA was demonstrated to be a promising tool for distinguishing high-grade gliomas from primary lymphoma and brain metastases. Additionally, the MRI characteristics of the NEPA were found to correlate with prognosis and treatment response. The purpose of this narrative review was to describe MRI features of the NEPA obtained with conventional and advanced MRI techniques to better understand their potential in identifying the different characteristics of high-grade gliomas, primary lymphoma and brain metastases and in predicting clinical outcome and response to surgery and chemo-irradiation. Diffusion and perfusion techniques, such as diffusion tensor imaging (DTI), diffusional kurtosis imaging (DKI), dynamic susceptibility contrast-enhanced (DSC) perfusion imaging, dynamic contrast-enhanced (DCE) perfusion imaging, arterial spin labeling (ASL), spectroscopy and amide proton transfer (APT), were the advanced MRI procedures we reviewed.

## 1. Introduction

Conventional and advanced MRI techniques have added valuable information in the diagnosis, characterization and prognosis of brain tumors [1]. Nevertheless, their application in the assessment of the enhancing part of the tumor has shown limitations in the achievement of sufficient accuracy for precise histological diagnosis and prognosis in the follow up [2,3]. Therefore, some authors have paid attention to the non-enhancing peritumoral area (NEPA), which is defined as the high-signal-intensity region in T2-weighted and fluid-attenuated inversion recovery (FLAIR) images surrounding tumoral lesions. The NEPA corresponds to a wide range of different pathological processes, including edema, and can involve local changes in vessels and cellular membrane permeability, peritumoral infiltration of neoplastic cells and necrosis, inflammation and gliosis [4,5]. The analysis of the NEPA has been demonstrated to be useful for the differential diagnosis of contrast-enhanced brain tumors, such as high-grade gliomas (HGGs), brain metastases (BMs) and primary central nervous system lymphomas (PCNSLs). Furthermore, the prognostic value of the extent and imaging characteristics of the NEPA from preoperative MRI has been explored, revealing a correlation with overall survival in patients with glioblastoma [6]. Additionally, the evaluation of the NEPA may help in the follow up of treated brain tumors in order to assess the presence of residual or recurrent tumors and in the differential diagnosis of tumor progression and treatment related changes; in particular, in HGGs, where the tumor infiltration is often beyond the contrast-enhanced area so that some authors suggest considering the NEPA as a non-enhancing portion of the tumor [7].

In this narrative review, the MRI features of NEPAs for differential diagnosis of contrast-enhanced brain lesions, prognostic evaluation and follow up of treated brain tumors are illustrated. The MRI characteristics of NEPAs are listed for both conventional MRI studies and advanced MRI techniques, including diffusion techniques, such as diffusion tensor imaging (DTI) and diffusional kurtosis imaging (DKI); perfusion-weighted imaging (PWI) techniques, such as dynamic susceptibility contrast-enhanced (DSC) perfusion imaging, dynamic contrast-enhanced (DCE) perfusion imaging and arterial spin labeling (ASL); spectroscopy; and amide proton transfer (APT).

## 2. Cellular and Histologic Characterization of the NEPA

A NEPA is defined as an area with high signal intensity in MRI sequences sensitive to elevated concentrations of fluids (T2-weighted and FLAIR sequences). Two kinds of NEPA can be identified: a non-infiltrative NEPA consisting of vasogenic edema (pure edema) and an infiltrative NEPA, also characterized by the presence of infiltrative neoplastic cells; for example, in HGGs. The NEPA observed in BMs parallels the increase in extracellular fluid, leading to severe vasogenic edema. In contrast, the infiltrative edema is one of the factors responsible for HGG recurrence and tumoral spreading after apparently complete surgical resection [8]. Engelhorn et al. found that MRI contrast-enhanced images underestimated glioma volume compared to fluorescence microscopic images and immunohistochemistry with surgical specimens. Indeed, the tumor can extend beyond the contrast-enhanced area, and the NEPA includes infiltrative neoplastic cells and glial alterations with an inflammatory microenvironment resulting from reactive astrocytes and microglia. Reactive swollen astrocytes are characterized by over-expression of water channel aquaporin-4, which is a cellular attempt to restore the intra- and extracellular balance of fluids altered by neurotoxic compounds secreted by malignant gliomas. The NEPA is also characterized by accumulation of activated abnormal microglial cells, with hyperexpression of CD11b and macrophage migration inhibitory factor (MIF) leading to an inflammatory microenvironment [9]. The main function of microglial cells is probably to defend the healthy brain against glioma cells but, on the other hand, they have been suspected to promote tumor growth through the release of angiogenic cytokines [10]. The infiltrative pattern of PCNSL differs from gliomas: PCNSL cells infiltrate the peritumoral area, growing along pre-existing vessels without true neoangiogenesis. Additionally, the peripheral infiltration of PCNSL probably causes reactive changes in the surrounding brain, leading to vasogenic edema as well [11].

## 3. Conventional MRI and Advanced Techniques in the Differential Diagnosis of HGGs, Lymphoma and BMs

### 3.1. Conventional MRI

The mass-edema index has been reported to be lower in HGGs than in BMs [12,13]. Elsewhere, the midline shift has been reported to be greater in primary brain tumors than in BMs, a midline shift > 5 mm being more common in primary tumors [12]. T2 hyperintensity involving the cortex but without contrast enhancement in the area adjacent to a solitary enhancing lesion has been reported as a frequent and relatively specific sign of glioma compared to BMs [14]. Similarly, the gyral involvement of a T2 hyperintensity peritumoral non-enhancing area was observed in 33 HGG cases and in none of 44 cases of PCNSL. This sign showed high specificity in differentiating HGGs from PCNSL (the sensitivity, specificity, positive predictive value and negative predictive value were 45.8%, 100%, 100% and 53.0%, respectively) [15]. An example of the differences between gyral involvement in HGGs and BMs is shown in Figure 1.

### 3.2. Diffusion Imaging Techniques

#### 3.2.1. Diffusion-Weighted Imaging (DWI)

Diffusion-weighted imaging (DWI) depicts the freedom movement of water molecules in tissues. A DWI sequence is based on the incomplete rephasing of water molecules that move in the interval between the initial dephasing (the 90° radiofrequency (RF)) gradient pulse and the second rephasing (the 180° RF) gradient pulse, resulting in a loss of signal [16]. The apparent diffusion constant (ADC) is the calculated diffusion constant and comprises a number of underlying diffusion environments (intracellular, extracellular and intravascular spaces) and processes [17]. In oncology, the ADC maps the cellularity density in the tumoral area of the central nervous system (CNS) lesion, as well as in the NEPA.

Several studies have found significant differences in ADC values in the enhanced tumor area and in the NEPA for HGGs, BMs and PCNSL [18,19,20,21]. Quantitative ADC analysis with DWI showed that HGGs have significantly higher ADCs in the enhanced tumor area and lower ADCs in the peritumoral edema area when compared with the corresponding tumoral and peritumoral ADC values in PCNSL. The combination of ADC values in the tumor necrosis, in the enhanced tumor area and in peritumoral edema can differentiate GBM from PCNSL with a sensitivity of 90% and specificity of 86% [22]. The appearance of the ADC pattern for the NEPA in a patient with PCNSL is shown in Figure 2.

The presence of various degrees of necrosis, hemorrhages and susceptibility artifacts in both HGGs and BMs is probably responsible for the lack of differences in the intra-tumoral ADC values between the two groups of enhancing lesions [18]. On the other hand, NEPA could be helpful for distinguishing between HGGs and BMs. Several studies have found that the ADC value for the peritumoral area in HGGs is lower than that in BMs [18,23]. The restricted water diffusion observed in the hyperintense area surrounding HGGs is probably due to neoplastic cell infiltration [23]. In infiltrative NEPAs, the presence of neoplastic cells mixed with areas of vasogenic edema hinders the free movement of water molecules, leading to lower ADC values than in pure vasogenic edemas [20]. Some studies have proposed ADC cutoff values ranging from 1.302 to 1.332 for the distinction between HGGs and BMs, with sensitivity and specificity, respectively, from 83 to 95% and 79 to 100% [23,24,25]. Han et al. found that higher b-values discriminated more accurately between HGGs and BMs, with sensitivity of 94.1% and specificity of 93.3% for a cutoff value of ADCmin = 0.890 × 10^−3^ mm^2^/s at a b-value of 3000 s/mm^2^ [24].

The appearance of infiltrative and vasogenic patterns for the NEPA in a patient with an HGG and BM is shown in Figure 3 and Figure 4, respectively.

The strength of DWI in distinguishing between enhancing CNS lesions is its availability in MRI centers: DWI is a straightforward imaging technique, it is not time-consuming, the postprocessing of data is very simple and it is routinely used even in centers that do not perform advanced imaging.

#### 3.2.2. Diffusion Tensor Imaging (DTI)

Diffusion tensor imaging (DTI) can be used to measure the complex anatomical structure of white matter. DTI consists of successive DWI acquisitions with encoding in various directions in order to depict the direction and the degree of anisotropic diffusion (diffusion in which one direction is preferred) and to enable the computation of the diffusion tensor. At least six directions of diffusion encoding are needed to calculate the diffusion tensor, which is usually described in terms of three coordinate axes in which the principal axis is the direction of preferred diffusion [26]. The main parameters obtained through the computation of a tensor model are the mean diffusivity (MD) and the fractional anisotropy (FA). MD is a measure of overall diffusion within a voxel; FA measures the directionality of water molecule movements and indirectly indicates the white matter fiber integrity and cell density.

DTI has been applied in the study of NEPAs with some contradictory findings. MD was reported to be higher and FA lower within the NEPAs of BMs when compared with NEPAs of HGGs. This can be explained by the more extensive and severe vasogenic edema surrounding cerebral metastases: the increase in extracellular fluid leads to the increase in the diffusion of water molecules in every direction (MD) and, at the same time, the decrease in the directionality (anisotropy) of water diffusion, resulting in a reduction in FA values [21]. Nevertheless, other authors reported lower FA values in NEPAs associated with HGGs than in NEPAs associated with BMs due to the tumoral infiltration of white fibers [27]. Figure 4 and Figure 5 illustrate DTI studies with patients with BMs and HGGs, respectively.

The combination of FA and MD measurements for NEPAs using cutoff values for FA > 0.22 and MD < 143 × 10^−6^ mm^2^ s^−1^ could differentiate HGGs from BMs with sensitivity and specificity of 90 and 100% if associated with a relative cerebral blood volume (rCBV) cutoff value > 3.14. In contrast, the same DTI parameter in the tumoral-enhancing area did not show significant differences between GBM and BMs [18].

#### 3.2.3. Diffusion Kurtosis Imaging (DKI)

Diffusion kurtosis imaging (DKI) is an advanced MRI method for the analysis of diffusion attempting to account for the non-Gaussian behavior of diffusion that can better characterize the in vivo effect of the complex cytoarchitecture of organic tissue formed by various compartments, cell types and intracellular constituents [28,29]. DWI and DTI are based on the simplified premise of the Gaussian distribution of water diffusion in tissues. Nevertheless, the biological complexity of brain tissue probably deviates from this simplified model. Therefore, DKI has been proposed as a better tool for detecting abnormalities in brain tissue than DTI [29]. Mean kurtosis (MK) has been demonstrated to parallel tissue complexity: MK is high in histologically complex lesions with elevated cellularity, cellular heterogeneity, necrosis, hemorrhage and vascular proliferation [30].

Both HGGs and PCNSLs are characterized by densely packed tumor cells and a small extracellular space [31]. Kurtosis parameters were higher in the enhanced parts of PCNSLs than in the solid enhanced parts of HGGs [19,32] because of the elevated cellularity and nuclear-to-cytoplasm ratio observed in PCNSLs, which are higher than in HGGs. Otherwise, kurtosis was higher in the NEPAs of the HGGs than in the PCNSLs, probably due to cellular infiltration of the NEPA in the case of the HGGs [32,33]. DKI has been reported to be more sensitive than DWI in detecting microstructural differences between these two tumor types [19].

DKI parameters for the tumoral solid part do not show significant differences between HGGs and solitary brain BMs; otherwise, DKI values for the NEPA are higher in HGGs than in solitary brain BMs [34]. Tan et al. found that neoplastic NEPA cells in HGGs were the cause of the increased restriction of diffusion both in the axial and radial directions [34].

### 3.3. Perfusion Imaging Techniques

PWI is an advanced MRI technique that allows the assessment of neoangiogenesis in tumors. The DSC and DCE perfusion imaging techniques require the use of a contrast agent, while ASL is a non-invasive tool that can be performed without a gadolinium injection.

DSC perfusion imaging is the most commonly used technique, and it relies on the measurements of the T2 MR signal during the first pass of a bolus of a paramagnetic contrast medium to provide maps of rCBV and relative cerebral blood flow (rCBF) [35,36]. The rCBV is a biomarker of neoangiogenesis and has been demonstrated to be directly correlated with tumor vascularity and grading [37,38,39,40,41]. In DCE T1-weighted MR perfusion imaging, rapid, serial T1 images are acquired during the administration of an intravenous contrast agent. DCE imaging is more reliable in quantifying vascular permeability, has higher spatial resolution and is less sensitive to susceptibility artifacts than DSC imaging. DCE MRI provides permeability parameters, such as the volume transfer constant (K-trans), the flux rate constant (Kep), the fractional blood plasma volume (Vp) and the fractional volume of the extravascular and extracellular space (Ve) [42]. Arterial spin labeling (ASL) allows the assessment of cerebral blood flow (CBF) in tumors by using magnetically labeled arterial blood water as an endogenous tracer [43]. Arterial blood external to the imaging section is labeled with an inversion pulse and, after a transit time, a scan is performed in the imaging section. The perfusion-weighted image is obtained from the subtraction of a control image without prior labeling, leaving the transported magnetization only [44]. CBF is the most commonly mentioned parameter in ASL [45]. One of the most practical advantages of the ASL technique is the immediate availability of perfusion images and the ease of quantification of CBF without extensive postprocessing.

PWI techniques have been applied in the differential diagnosis of solitary enhancing lesions: HGGs, BMs and PCNSLs. Results from previous studies on PWI analyses in solid enhancing tumor areas of HGGs and BMs were not concordant: some studies found no significant differences in PWI values between the two [46,47,48,49], while others demonstrated that PWI parameters could distinguish HGGs from BMs [18,50].

#### 3.3.1. Dynamic Susceptibility Contrast-Enhanced Perfusion

Analyses of MR perfusion patterns in the NEPA have shown more coherent results than those of the perfusion patterns for the enhancing tumors in the differential diagnosis of HGGs and BMs. DSC perfusion studies demonstrated that rCBV in the NEPA is higher for glioblastomas than for BMs [18,46,47,48,49,51,52], and that its performance in discriminating HGGs and BMs is better than that of the rCBV in the solid part of the tumor, with different diagnostic accuracies reported in the literature. Askaner et al. found areas under the curve (AUCs) of 0.74 and 0.68, respectively, with normalized rCBV threshold values of 1.56 for NEPA and 3.75 for the enhancing region [48], while Neska-Matuszewska et al. reported a higher accuracy of 0.94 for a max rCBV cutoff value of 0.98 in the peritumoral zone [47] and no significant differences for the max rCBV within the enhancing tumor. A positive predicting value of 68% and a negative predicting value of 95% for normalized rCBV (cutoff value of 1) in the NEPA was reported by Blasel et al., suggesting than BMs can be reliably excluded if the rCBV is increased in the NEPA while, conversely, a decreased rCBV may not exclude HGGs [53]. These findings probably reflect the different histological characteristics of NEPAs in HGGs and BMs: the hypoperfusion of peritumoral edema in BMs is the result of vasogenic edema due to plasma fluid leakage and local compression of the microcirculation caused by the edema itself [54,55], while the moderate hyperperfusion reported in the non-enhancing area around HGGs is the consequence of infiltrative neoangiogenesis and increased neoplastic cellularity [56]. Moreover, the direct relationship between perfusional values and tumor cell density in the peritumoral T2 hyperintensity region has been demonstrated in murine models of glioblastomas [57]. In contrast, the solid enhancing tumor part is probably more similar in HGGs and BMs, both presenting angiogenesis and blood–brain barrier breakdown [58]. The patterns of PWI in NEPAs in HGG and BM patients are shown in Figure 6 and Figure 7, respectively.

To evaluate the hypothesis of tumor infiltration beyond the NEPA, the CBV values outside—but within the region adjacent to—the peritumoral T2 hyperintensity area were assessed but did not show differences in HGGs compared to BMs [48,59].

In the differential diagnosis of PCNSL and other enhancing tumors, such as HGGs and BMs, a major role for the tumor core compared to the peritumoral area has been demonstrated. Neska-Matuszewska et al. proposed a two-step approach for differentiating HHGs, BMs and PCNSLs that involved placing small ROIs over several hotspots in the enhancing core of the solid tumor and the NEPA and choosing the highest rCBV value from all ROIs as the max rCBV. They used the max rCBV (max rCBV > 2.18) assessed in the tumor core to differentiate PCNSLs, which are hypoperfused, from HGGs and BMs, both hyperperfused, with an accuracy of 0.98 and the max rCBV (max rCBV above 0.98) measured in the NEPA to identify the tumor infiltration, present in HGGs but not in BMs, with an accuracy of 0.94 [47]. Hypoperfusion in PCNSLs can be explained by hypovascularization, increased vascular permeability and the absence of neoangiogenesis [60], whereas neovascularization is a hallmark of HGGs with variable vascular permeability [61,62]. Attention should be paid to the preloading of the bolus of contrast media to correct CBV when performing PWI studies of lymphomas because gadolinium extravasation into the interstitial space is very high in lymphomas and may differentiate them from HGGs [2]. Figure 2 and Figure 6 show the appearance of PCNSL and HGG tumors and NEPAs in DSC imaging studies.

#### 3.3.2. Dynamic Contrast-Enhanced Perfusion Imaging

DCE imaging provides a measure of vascular permeability, K-trans and Vp being the commonest parameters assessed. Very few studies have explored the NEPA with DCE imaging. K-trans measured with a 3-tesla scanner in the peritumoral area did not differ in HGGs and in BMs [18], probably reflecting the absence of blood–brain barrier breakdown in this area. Nevertheless, another study proposed DCE imaging for the distinction between tumoral and non-tumoral T2 hyperintense non-enhancing areas in patients with HGGs and BMs [59]. Higher values for Vp and K-trans were demonstrated in tumoral NEPAs compared to non-tumoral NEPAs. The higher plasma volume (Vp) and tissue permeability (K-trans) found in tumoral NEPAs possibly derive from increased vessel density, abnormally structured tumoral blood vessels and impaired blood–brain barrier as a consequence of neoangiogenesis. Additionally, the same study showed that Vp and K-trans are not significantly different in non-tumoral NEPAs, as observed in BM patients, and in the normal-appearing white matter. An example of a tumoral NEPA from a DCE imaging study is shown in Figure 8.

#### 3.3.3. Arterial Spin Labeling

Previous studies have indicated that the CBF derived from ASL shows a good correlation with the CBV, the CBF derived from DSC MRI and histopathological vascular density. Additionally, ALS is less affected by a disrupted blood-brain barrier and by the underestimation of CBV values due to the leakage of gadolinium [43,63]. Arterial spin labeling (ASL) allows the assessment of cerebral blood flow (CBF) in tumors by using magnetically labeled arterial blood water as an endogenous tracer [43]. ASL has been applied in distinguishing HGGs from PCNSLs and BMs. Both the intratumoral and the peritumoral CBFs were found to be significantly higher in glioblastomas than in metastases, with AUCs of 0.71 and 0.83, respectively [64]. Weber et al. confirmed these findings with a threshold of 0.5 for the CBF within the peritumoral NEPA region, providing sensitivity, specificity, PPV and NPV of 100%, 71%, 94% and 100%, respectively. Additionally, the same study found that the CBF and arterial transit time (ATT) in the peritumoral area were higher in HGGs than in PCNLs, with an AUC of 0.96 and sensitivity, specificity, PPV and NPV comparable to tumoral ROIs. You et al. found significantly lower intratumoral peritumoral CBF values in PCNSLs than in HGGs, with better diagnostic performance for the peritumoral CBF (AUC > 0.90) than intratumoral CBF (AUC > 0.80) [65].

In contrast, Lin et al. found no significant differences between tumoral CBFs in HGGs and BMs, while the peritumoral CBF was higher in glioblastomas. Interestingly, they explored the gradient of CBF values within the peritumoral T2 hyperintense area from the region close to the tumor to the region distant from the tumor border, showing that the CBF gradient was able to discriminate HGGs and BMs well with a cutoff value of 1.92 mL/100 g, sensitivity of 92.86% and specificity of 100.00% [66]. The steeper CBF gradient observed in HGGs reflects the gradient of tumoral cell infiltration, which decreases from the central to peripheral T2 hyperintense area [67,68]. Additionally, the CBF gradient in the peritumoral area was more accurate than the ADC gradient in discriminating HGGs from BMs. Figure 9 displays a case of BM with low values for the CBF in the NEPA.

### 3.4. H-Magnetic Resonance Spectroscopy

H-Magnetic resonance spectroscopy (H-MRS) is a non-invasive technique that provides information about the metabolic characteristics of tissue in the brain [69,70]. H-MRS uses the hydrogen (H) nucleus to obtain the MR spectra. The spectrum is acquired for a volume of interest (VOI) positioned on anatomical images in order to select the structure of interest. A spectroscopy study can be performed on a single- or multi-voxel basis using both long and short echo times (ETs). Two techniques can be used for 1H-MRS: point-resolved spectroscopy (PRESS) and the stimulated echo acquisition mode (STEAM) [71].

Choline (Cho) is an index of cellular proliferation and membrane turnover. Creatine (Cr) is present in metabolically active tissue and is used as a standard for calculating ratios because its peak is relatively constant. Increases in the Cho/Cr peak within tumors have been described in the NEPAs of HGGs, indicating tumor cells’ presence beyond the area with contrast enhancement, but not in the peritumoral edema of BMs [51,72,73,74]. Further, Tsougos et al. found that a decrease in the N-acetylaspartate (NAA)/Cr ratio, in addition to an increase in Cho/Cr, differentiated HGGs from BMs [51]. Additionally, the myoinositol (mI) and glycine (Gly) peaks have been found to increase in the peritumoral region in HGGs and not in BMs [75]. The increased Cho/NAA and mI + Gly/NAA ratios might also be explained by a decrease in NAA. NAA is a neuronal marker that can decrease in the NEPA in HGGs because of the infiltration of neoplastic cells and the consequent impairment of brain tissue. In contrast, the pattern of growth of metastases probably preserves the integrity of the brain tissue around the tumor borders. Myoinositol is involved in glial proliferation [76] and was detected not only in NEPAs but also in the normal contralateral white matter in cases of HGGs [77]. It should be noted that the peaks of mI and glycine can be better assessed using short-echo-time MRS [69]. According to Chawla et al., the lipid + lactate/Cr peak did not help in the differential diagnosis of HGGs and BMs, whereas higher lipid + lactate/Cr levels were demonstrated in the NEPAs of PCNSLs compared to HGGs and BMs [78]. Furthermore, Bendini et al. found that the combination of a high Cho/Cr ratio and the presence of lipid and lactate peaks in the NEPA allowed the differential diagnosis of HGGs, BMs and PCNSLs [73]. Figure 10 and Figure 11 depict the differences between MRS studies of HGGs and BMs.

The NEPA was further characterized using high-field-strength 3-tesla MRI, which allows a better signal-to-noise ratio and disclosure of more metabolites [79,80]. Beyond the Cho/Cr ratio, other metabolites have also been assessed in the NEPA in HGGs and BMs. In both groups, an increased glutamate plus glutamine-to-creatine (Glx/Cr) peak, as well as those for aliphatic amino acids, such as valine, leucine and isoleucine, was detected. The increase in Glx/Cr may have been the consequence of the increased activity of the glutamine synthetase enzyme that astrocytes synthesize by removing excess glutamate. Additionally, the accumulation of Glx may have a toxic effect on myelin structure, leading to a mobilization of membrane phospholipids, which would account for the presence of amino acid and lipid peaks [81,82].

It has also been observed that the Cho/Cr peak in the peritumoral area may be affected by steroids: a reduction in the Cho/Cr peak was observed after the use of dexamethasone in low-grade gliomas (LGGs), but the changes were not significant [83].

### 3.5. Amide Proton Transfer

Amide proton transfer (APT) imaging is sensitive to the presence of amide protons in mobile proteins and peptides. APT is a chemical exchange saturation transfer (CEST) imaging technique that is based on the saturation transfer from mobile proteins and peptides to water by means of the transfer of saturated protons of the amide groups [84]. Mobile proteins are the secreted proteins and are also found in the cytosol and in the endoplasmic reticulum. In brain tumors, a close relationship between the proliferative activity of the tumor and mobile protein synthesis is assumed. As a consequence, increased protein synthesis leads to an increased concentration of cytosolic proteins contributing to the APT signal, as shown in animal models of gliosarcomas [85]. A correlation between the APT signal and glioma grade has been demonstrated [86]. APT imaging takes advantage of higher magnetic fields (3-tesla) and the recent development of 3D sequences allowing coverage of a large part of the brain in approximately 7–8 min [87].

An increased APT signal may derive from contributors other than the proliferative rate of the tumor. A high APT signal has been observed in tumoral cystic components [88] independently of the grade. Additionally, increased APT signals have also been reported in blood vessels [89] and, as a consequence, in hypervascular tumors. Nevertheless, acute hemorrhages also show increased APT signals, even in the absence of increased cellularity [90]. Therefore, APT in brain tumors should be interpreted with caution.

Wen et al. explored the peritumoral area and found that both the immediate edema (adjacent to but distinct from the core of the tumor, within a 1 cm margin) and the peripheral edema (outside the immediate edema regions) had lower APT signals than the tumor core and that of the peripheral edema was slightly lower than the immediate edema. The APT signal in the peritumoral area was higher than in the contralateral normal-appearing white matter, probably because intravascular proteins and peptides enter the extravascular space as a consequence of the increased pressure gradient from the intravascular to extravascular space due to proximity to the tumor [91].

Additionally, absolute APT parameters (max, min, mean) and APT values normalized for the values of the normal-appearing white matter were shown to be lower in the peritumoral edema of BMs than HGGs, with the minimum APT signal being the most accurate parameter in the differential diagnosis [91]. In HGGs, tumoral and inflammatory cells infiltrate into the extravascular space through a leaky blood–brain barrier, contributing to a higher content of mobile peptides and proteins in the peritumoral areas of HGGs compared to BMs [91]. Figure 12 shows the different APT signals in the peritumoral area compared to the surrounding normal-appearing white matter, suggesting an infiltrative pattern for NEPAs in HGGs.

### 3.6. Multiparametric Approaches, Diagnostic Algorithms and Tumor Segmentation

Several studies have reported that the best diagnostic performance is achieved when combining different advanced MRI techniques. Bauer et al. reported a high accuracy (AUC value of 0.98) for the association between rCBV, FA and MD measured in NEPAs in the differential diagnosis of HGGs and BMs [18]. Additionally, the combination of DTI and PWI based on tumoral ADC and FA values and peritumoral rCBV values was found to be an optimal model to distinguish HGG from non-HGG lesions (PCNSLs and BMs), with an AUC = 0.938 [92]. When comparing the diagnostic performances of MR advanced techniques in differentiating BMs from HGGs, DSC PWI had better diagnostic performance than MR spectroscopy and ADC measurements when assessed in the peritumoral region [47,93]. Figure 4, Figure 5, Figure 6 and Figure 7 show MD and CBV maps for NEPAs in cases of BMs and HGGs. These findings were confirmed by Toh et al., who reported that peritumoral rCBV and NAA/Cr, Cho/Cr and Cho/NAA ratios could better discriminate glioblastomas from intracranial metastases than ADC and FA values [50]. In Figure 10 and Figure 11, the MRS peaks in the NEPAs in two cases (an HGG and BM) are displayed.

Qualitative diagnostic algorithms and radiomic approaches have been proposed for the differential diagnosis of BMs and HGGs [18,25,94,95,96,97]. Radiomics consists of the conversion of radiological images into quantitative data with the aim of helping clinical decision making by enhancing the data available.

Voicu et al. compared four algorithms evaluating both conventional qualitative MRI features and quantitative analyses of DWI and DSC PWI parameters in NEPAs. The study confirmed that, among the quantitative parameters, the mean-normalized relative CBV and the mean CBV percentage recovery in NEPAs were more accurate than the ADC in the discrimination of HGGs from BMs, with cutoffs of 1.46 (sensitivity: 83%, specificity: 89%, AUC: 0.904) and 87.5% (sensitivity: 88.9%, specificity: 94.4%, AUC: 0.907) [98]. The quantitative algorithm based on PWI and DWI values had better diagnostic performance than the qualitative approaches, where the radiologists were asked to give the diagnosis solely on the basis of morphological information, and the semi-quantitative algorithm, where radiologists were also helped in the diagnosis by PWI and ADC values based on cutoffs derived from the literature. Additionally, the quantitative algorithm was the algorithm with the best concordance with histologic findings [98].

The application of automated MRI pattern recognition techniques, such as support vector machine (SVM) algorithms, confirmed that the combination of advanced MRI techniques—in particular, those employing the decrease in NAA/Cr and the increase in rCBV in the peritumoral area—allowed the differentiation of glioblastomas versus metastases. Interestingly, the successive inclusion of the lipids and lactate (Lip + Lac)/Cr ratio for the NEPA in the algorithm as an additional feature decreased the performance of this approach, probably because of the absence of necrosis in this area, as confirmed by Chawla et al., who showed that peritumoral lipids and lactate did not help in the differential diagnosis of HGGs and BMs [74,78].

A radiomic approach based on multiparametric conventional MRI (T1, T2, FLAIR and T1 with gadolinium) allowed the classification of NEPAs into tumoral and nontumoral components. The machine learning algorithms were trained, tested and validated with data from NEPAs in patients with HGGs and BMs. The plasma volume (Vp) and tissue permeability (K-trans) derived from the DCE PWI were used for validation of the classification results based on the consistency with the literature data, with higher Vp and K-trans reported in tumoral NEPAs than in non-tumoral NEPAs. As expected, the radiomic analysis categorized the entire NEPA as non-tumoral in BM patients and as a mixed pattern of tumoral and non-tumoral components in HGG patients. Interestingly, in HGG patients, tumoral features were found within the NEPA adjacent to the enhancing lesion. The main limitation of this study was the absence of histological confirmation of the validation results [59].

The parameters derived from advanced MRI techniques may also be assessed using segmentation of the tumor core and the edema. The manual segmentation of brain tumors in structural MRI images is time-consuming, laborious and needs the expertise of a neuroradiologist. Efforts have been made in achieving automatic and robust tumor segmentation. Additionally, the differences between the NEPA and the enhancing part of the tumor hinder the performance of the segmentation in MRI images. Several researchers have been tempted to employ approaches that segment the boundary of the tumor core and the NEPA, the deep learning methods being the most effective [99,100]. Additionally, Ranjbarzadeh et al. proposed a pre-processing strategy to overcome the difficulties affecting deep learning methods in dealing with the imbalanced data between small-sized tumors and the rest of the brain. The pre-processing strategy consisted of removing the unimportant information from the MR images, allowing the use of a simpler deep learning model to segment the NEPA and the enhancing part of the tumor. The segmentation was based on the convolutional neural network (CNN) model and used four MRI modalities, (FLAIR, T2-weighted and T1-weighted without and with gadolinium), and it reached performance close to that of human observers [101].

## 4. Conventional and Advanced MRI Techniques in the Assessment of NEPAs for Prognosis and Treatment Response

### 4.1. Correlation between NEPAs and Prognosis

The association between prognosis and the extension of NEPAs in HGG patients has been reported in some studies [6,102,103,104], while it was not confirmed in others [105,106,107,108].

Schoenegger et al. found that extensive NEPAs in preoperative MRI correlated with poor prognosis independently of the postoperative Karnofsky performance score (KPS), age and type of tumor resection [6]. These findings were confirmed by Mummareddy et al., who showed that higher preoperative NEPA volume in 210 HGG patients directly correlated with decreased survival [109]. Intriguingly, the extent of the NEPAs was related to VEGF expression [110], suggesting a role for neovascularization in the development of the peritumoral area and, therefore, explaining the correlation between the extent of NEPAs and the prognosis of HGG patients [111]. The extension of NEPAs was recognized as an independent marker of early recurrence in HGG patients, along with the number of tumors in the case of multifocal HGGs and IDH mutation status [112], and as a prognostic factor for survival after RT at a conventional dose [6,113] or postoperative high-dose proton beam therapy [114]. Limited NEPAs (defined as an extent of peritumoral edema < 3 cm) were associated with more favorable survival and less tumor spreading compared to extensive NEPAs [114].

In addition, the role of NEPAs in predicting early HGG recurrence and prognosis was confirmed not only in preoperative but also in postoperative MR scans. An MRS study demonstrated that a higher Cho-NAA radio (Cho/NAA ≥ 1.31) measured in NEPAs in postoperative MRI was an independent risk factor for early recurrence. A prognostic scoring scale for GBM recurrence, including MGMT promoter methylation status, the radiation therapy and the Cho-NAA ratio in the NEPA, was proposed in order to divide patients into low-risk, moderate-risk and high-risk groups, with significant differences in the survival rates between groups [115]. Moreover, MRS was useful for a better definition of tumor margins, favoring a more precise identification of the radiation target within NEPAs and limiting the radiation exposure to the surrounding healthy brain [116]. Figure 10 shows the increase in the Cho/Cr ratio in an infiltrative NEPA in a case of HGG.

Tumor localization and tumor necrosis may influence the size of NEPAs and should be taken into account in the prognostic stratification of HGG patients: lobar superficial tumors and tumors with extensive necrosis were reported to have larger NEPAs [111,117], while deep tumors showed small amounts of edema, probably because denser fibers, such as the corpus callosum and the corticospinal tracts, prevent edema spreading compared to superficial fibers [118]. Moreover, surgical resection of deep tumors is more difficult than with superficial ones. These observations may explain the inconsistency in the results concerning the relationships between the extent of NEPAs and prognosis in different studies when the tumor location is not considered [117].

In patients with BMs, the ratio between the peritumoral edema and the tumor volume, but not the edema volume alone, was recognized as a prognostic factor for overall survival (OS) independent of other predictive factors, such as complete resection and postoperative brain irradiation [119].

Advanced MRI techniques and multimodal MRI may assist in the in vivo assessment of tumor infiltration and the prediction of future early recurrence and survival. Multiparametric pattern analysis from clinical MRI sequences using radiomic signatures determined via machine learning methods has allowed the estimation of tumoral infiltration in NEPAs and the prediction of the rate and locations of tumor recurrence in HGG patients. The recurrent tumor region showed higher vascularity and cellularity as a radiomic signature [120,121,122,123,124]. Moreover, perfusion parameters for HGGs derived from DSC imaging have been used to determine four types of tumoral and peritumoral vascular heterogeneity based on CBV and CBF values: high-angiogenic and low-angiogenic regions of the enhancing tumor, potentially tumor-infiltrated peripheral edema and vasogenic edema. The prognostic potential of the hemodynamic tissue signature for the prediction of patient survival was demonstrated by the longer survival times of patients with lower preoperative perfusion indexes for both tumors and NEPAs [125]. Furthermore, artificial intelligence (AI) has been applied to assess the heterogeneity of NEPAs in BMs and HGGs in order to extract a voxel-based deep learning AI index that was used to stratify HGG patients in terms of survival and IDH1 mutation status on the basis of DTI-based free water volume fraction maps [126].

### 4.2. Correlation between NEPAs and Treatment Response

Multiparametric MRI, including conventional, diffusion and perfusion imaging methods, has been applied to provide a more accurate analysis of NEPAs in terms of therapy response to bevacizumab treatment for recurrence in HGG patients [127]. The infiltrative tumoral area within the NEPAs was distinguished from vasogenic edema by means of PWI since high CBV, CBF, K-trans and Vp values were associated with tumoral infiltration. Throughout the bevacizumab therapy, the increase in the tumor-infiltrative areas in the NEPAs occurred even in the absence of the growth of the enhancing tumor volume. These findings are consistent with previous studies suggesting that anti-angiogenic therapy does not prevent the HGG infiltrative tumor progression [128] that can occur as an invasive, non-enhancing phenotype indistinguishable in conventional MRI from vasogenic edematous changes [129]. However, the volumes of tumor-infiltrative edema at weeks 8 and 16 after the bevacizumab treatment initiation were correlated with the progression-free survival, suggesting a prognostic role for NEPAs in predicting the response to bevacizumab treatment [127].

The prognostic role of NEPAs in treatment response was also explored in patients with BMs. MRI features of the peritumoral areas of BMs were proposed as a new biomarker for assessing T790M mutation, a secondary mutation in the EGFR gene responsible for the resistance to therapy with tyrosine kinase inhibitors (TKIs) of the EGF receptor (EGFR) in NSCLS patients. The radiomic signatures of pre-treatment BM NEPAs, together with radiomic signatures in the tumoral area, were strongly associated with the T790M mutation status [130].

Lastly, the extent and MRI features of NEPAs were investigated to distinguish radionecrosis from progression in BM patients. The extent of NEPAs was proposed as a potential marker of radionecrosis in patients with BMs treated with stereotactic radiosurgery: a high edema/lesion volume ratio was found to be predictive of radionecrosis as opposed to tumor recurrence [131]. Weis et al. proposed an imaging-driven biophysical model of NEPAs based on diffusion parameters, such as the cellular density marker, and tumoral growth from serial imaging time points, such as the proliferation rate index, to distinguish radionecrosis from tumor progression in BM patients [132].

## 5. Limits and Future Perspectives

Despite the potential of conventional and advanced MRI techniques in the assessment of NEPAs in the differential diagnosis of brain tumors, some limits of the abovementioned studies should be acknowledged.

These studies used different imaging acquisition parameters and forms of post-processing and analysis. Specifically, various approaches were used in the delineation of NEPAs. Some studies used manually drawn ROIs, which are poorly reproducible, while others utilized threshold-based methods. The proximity of ROIs to the tumor boundaries may also affect the results of NEPA assessment. In addition, the cutoff values previously proposed showed large overlaps between the histological subgroups and were not validated by independent external assessments or histological data. Additionally, the studies were mainly retrospective and monocentric and included small patient cohorts, with a consequent bias due to different patient selection and inclusion criteria.

Additionally, other MRI techniques have rarely been used in the assessment of NEPAs and may be of interest in future studies. For example, T2 mapping/relaxometry may help in distinguishing infiltrative from non-infiltrative NEPAs: T2 relaxation time is reduced in infiltrative NEPAs compared to infiltrative NEPAs due to the presence of malignant cells. Furthermore, phase difference-enhanced imaging (PADRE) selectively enhances both paramagnetic and diamagnetic substances and allows the visualization of myelin-rich structures and veins, and it may help in the assessment of the integrity of white matter structures and vascularity in NEPAs. Nevertheless, despite their potential interest in this field, comparisons of the two kinds of NEPAs with these two techniques have scarcely been explored [133,134].

Future large prospective studies aiming at the standardization of MRI acquisition protocols, as recommended by imaging societies [135,136], are warranted to define the actual role of NEPAs in brain tumors. More importantly, rigorous validation between centers and confirmation with clinical and histological data—using, for example, a neuro-navigation system for serial biopsy sampling—should be undertaken [52]. At present, NEPA assessment using advanced MRI techniques is unquestionably helpful in the imaging study of brain tumors and represents a promising tool for making NEPAs a robust quantitative imaging biomarker for differential diagnosis in brain tumors.

## 6. Conclusions

The study of NEPAs with conventional and advanced MR imaging techniques was shown to be helpful in discriminating HGGs, BMs and PCNSLs, as well as in predicting prognosis and treatment response, overcoming the diagnostic limits shown by the analysis of the tumor contrast-enhancing area. Advanced MRI techniques, such as PWI, DWI and MRS, are useful to distinguish infiltrative and non-infiltrative NEPAs, with infiltrative edema mainly demonstrating an increase in the perfusion parameters and Cho/Cr peak and a decrease in the NAA/Cr peak and ADC values. The accuracy of analyses increases when all these techniques are used in combination. Therefore, in the future, the radiomic evaluation of NEPAs based on multiple MRI parameters could be very useful in tumor assessment, along with large multicenter prospective studies including rigorous validation between centers and confirmation with histological data.

## Figures and Tables

**Figure 1 cancers-15-02992-f001:**
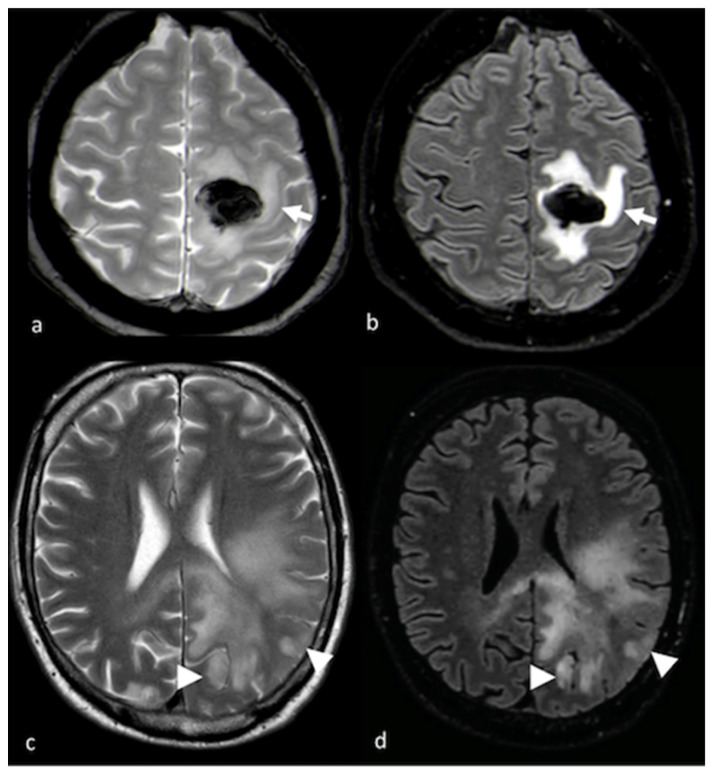
Cortical involvement in BMs and HGGs: (**a**,**c**) axial T2 GE-weighted image; (**b**,**d**) axial FLAIR T2-weighted image showing, respectively, a left frontal BM without T2 hyperintensity gyral involvement ((**a**,**b**) white arrows) and a parietal HGG with T2 hyperintensity cortical involvement ((**c**,**d**) arrowheads). BM: brain metastasis; HGG: high-grade glioma.

**Figure 2 cancers-15-02992-f002:**
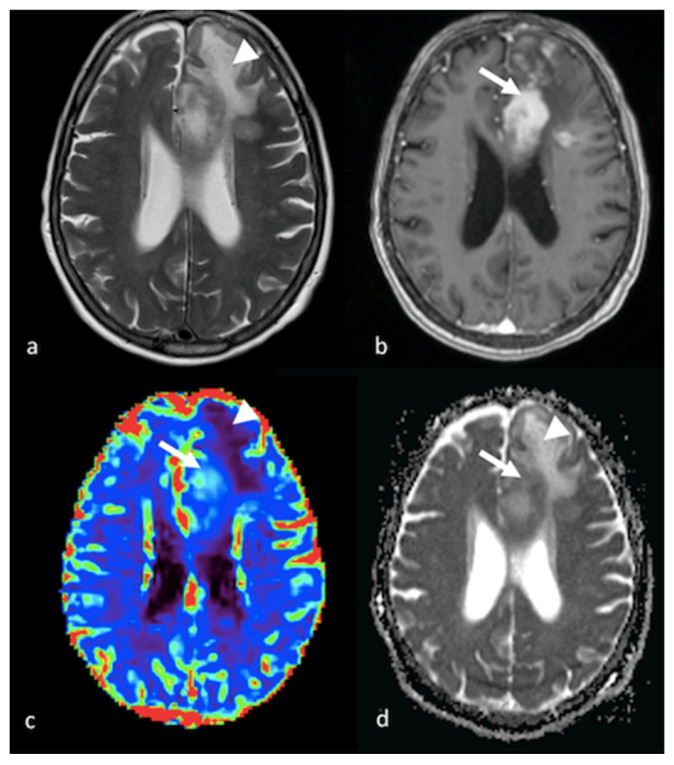
DWI and PWI in left frontal PCNSL: (**a**) axial T2-weighted image; (**b**) axial postcontrast T1-weighted image; (**c**) CBV map from DSC imaging study; and (**d**) ADC map from DWI study showing a left frontal lesion with intense enhancement and a large NEPA with low ADC and a mild increase in CBV within the contrast-enhanced lesion (white arrows) corresponding to elevated cellularity, high ADC values and hypoperfusion in the NEPA (white arrowheads), suggesting vasogenic edema. PWI: perfusion-weighted imaging; DWI: diffusion-weighted imaging; PCNSL: primary central nervous system lymphoma; CBV: cerebral blood volume; DSC: dynamic susceptibility contrast-enhanced; ADC: apparent diffusion coefficient; NEPA: non-enhancing peritumoral area.

**Figure 3 cancers-15-02992-f003:**
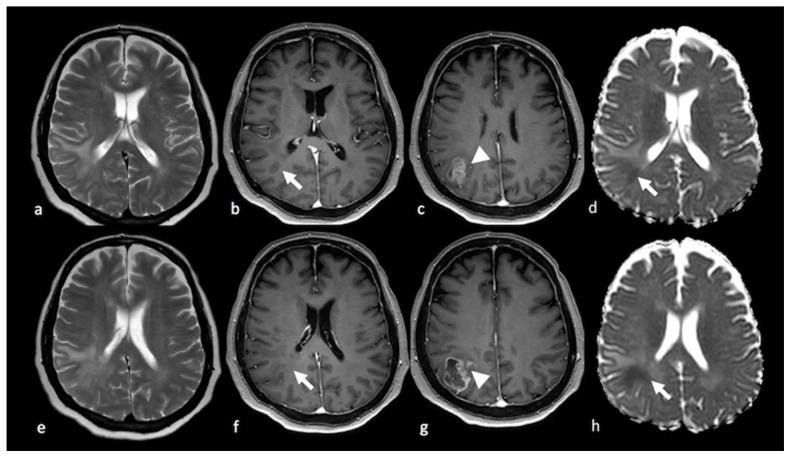
DWI of right parietal HGG: (**a**,**e**) axial T2-weighted image (**b**,**c**,**f**,**g**); axial postcontrast T1-weighted image; and (**d**,**h**) ADC map from DWI study showing a right parietal lesion with peripheral (ring) enhancement and NEPA with low ADC (white arrows) outside the contrast-enhanced lesion’s border (arrowhead), suggesting an infiltrative pattern for the HGG. DWI: diffusion-weighted imaging; HGG: high-grade glioma; ADC: apparent diffusion coefficient; NEPA: non-enhancing peritumoral area.

**Figure 4 cancers-15-02992-f004:**
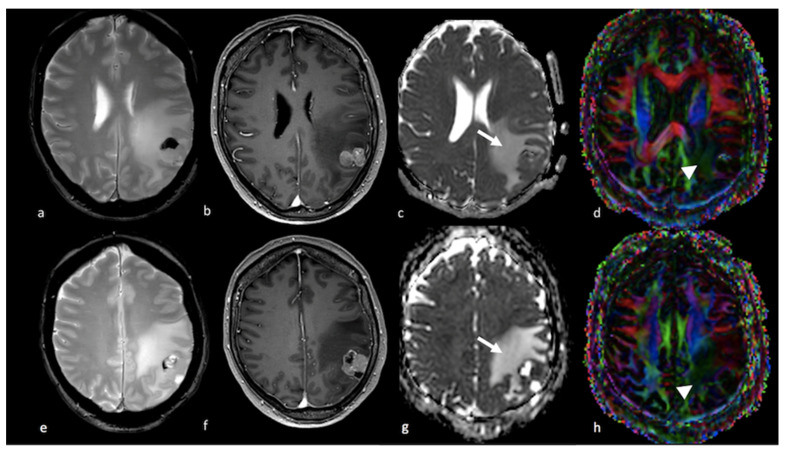
Diffusion imaging in left parietal metastasis: (**a**,**e**) axial GE T2-weighted image; (**b**,**f**) axial postcontrast T1-weighted image; (**c**,**g**) ADC map from DWI study; (**d**,**h**) DTI image displaying a lesion in the left parietal lobe characterized by heterogeneous contrast enhancement and the presence of blood products and showing an extensive NEPA. The NEPA is characterized by high ADC values (white arrow) due to vasogenic edema. The DTI FA color map shows a decrease in FA values due to the increase in extracellular fluid, which reduces the directionality of diffusion, leading to a reduction in the visualization of the white matter fibers (arrowhead). GE: gradient echo; ADC: apparent diffusion coefficient; DWI: diffusion-weighted imaging; DTI: diffusion tensor imaging; NEPA: non-enhancing peritumoral area; FA: fractional anisotropy.

**Figure 5 cancers-15-02992-f005:**
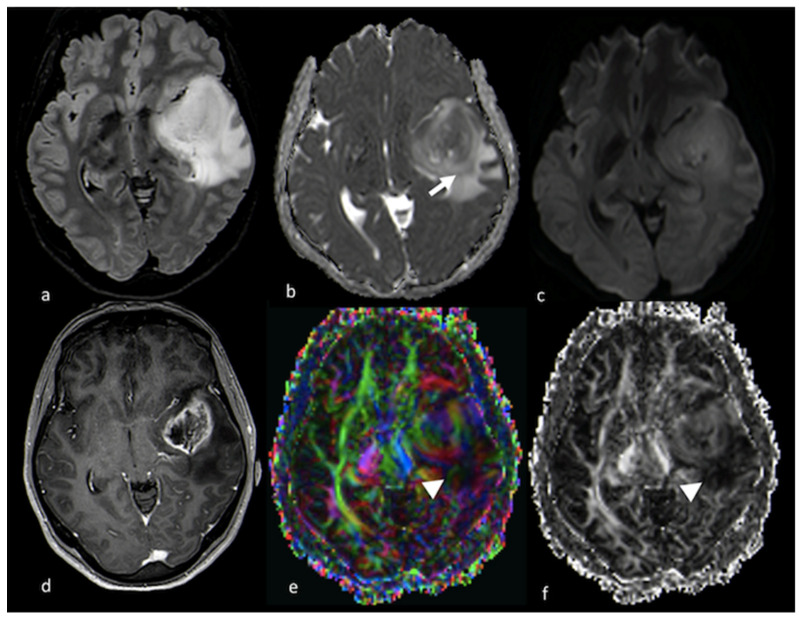
Diffusion imaging in left temporo-insular HGG: (**a**) axial FLAIR T2-weighted image; (**b**) ADC map from DWI study; (**c**) isotropic DWI images; (**d**) axial postcontrast T1-weighted image; (**e**) DTI FA color map; and (**f**) FA map from DTI study showing a left parieto-insular lesion with low ADC that extends over the contrast-enhanced rim (white arrow). The DTI study shows a decrease in FA values in the NEPA due to tumoral infiltration of white matter fibers (arrowhead). HGG: high-grade glioma; ADC: apparent diffusion coefficient; DWI: diffusion-weighted imaging; DTI: diffusion tensor imaging; FA: fractional anisotropy; NEPA: non-enhancing peritumoral area.

**Figure 6 cancers-15-02992-f006:**
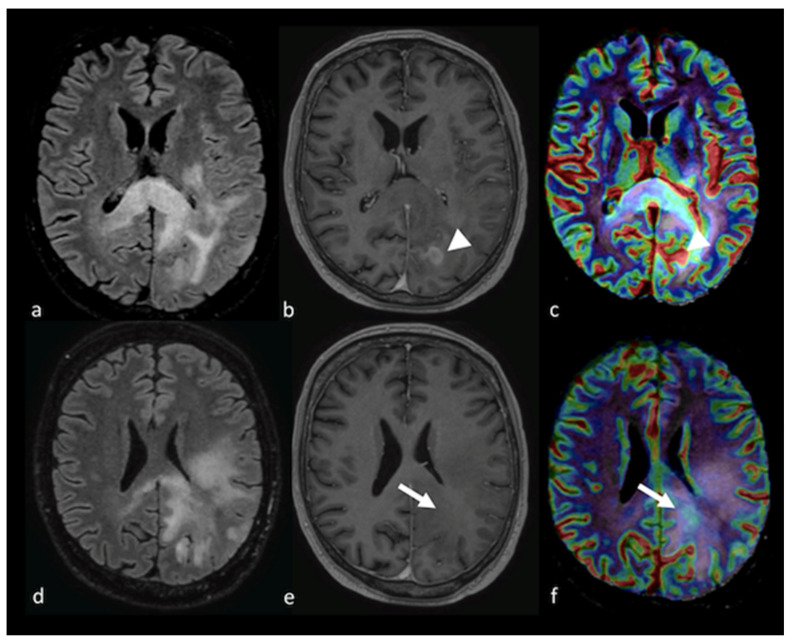
PWI in left parieto-occipital HGG: (**a**,**d**) axial T2 FLAIR image; (**b**,**e**) axial postcontrast T1-weighted image; and (**c**,**f**) CBV map from DSC imaging study showing an extensive T2 FLAIR hyperintense region affecting the left parietal and occipital lobe with the involvement of the corpus callosum and a ring enhancement component (arrowhead). The CBV map shows a mild increase in not only the contrast-enhanced area (arrowhead) but also in the NEPA (white arrow), suggesting an infiltrative pattern. PWI: perfusion-weighted imaging; HGG: high-grade glioma; CBV: cerebral blood volume; DSC: dynamic susceptibility contrast-enhanced; NEPA: non-enhancing peritumoral area.

**Figure 7 cancers-15-02992-f007:**
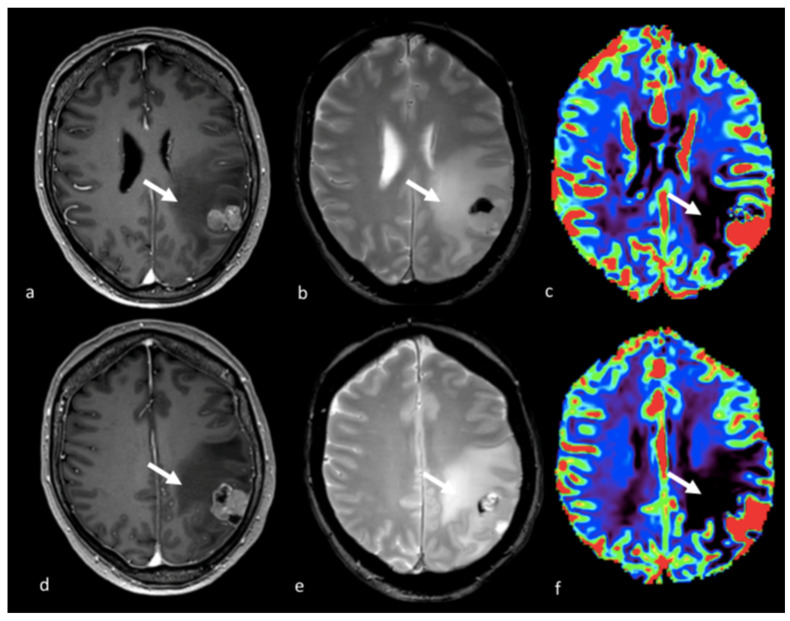
PWI imaging in left parietal metastasis: (**a**,**d**) axial postcontrast T1-weighted image; (**b**,**e**) axial GE T2-weighted image; and (**c**,**f**) CBV map from DSC imaging study showing a left parietal lesion with heterogeneous contrast enhancement and the presence of blood products with an extensive NEPA characterized by a very low CBV value (white arrow). This case is also shown in Figure 4. GE: gradient echo; CBV: cerebral blood volume; DSC: dynamic susceptibility contrast-enhanced; NEPA: non-enhancing peritumoral area.

**Figure 8 cancers-15-02992-f008:**
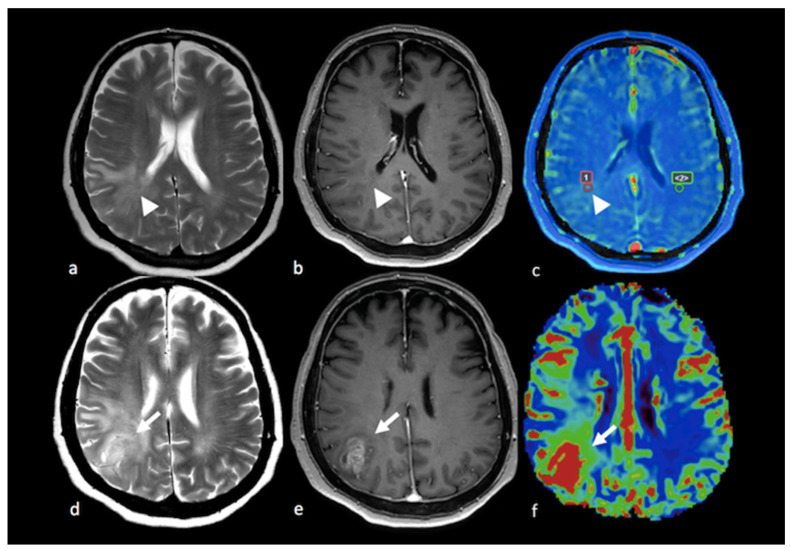
PWI in right parietal HGG: (**a**,**d**) axial T2-weighted image; (**b**,**e**) axial postcontrast T-1 weighted image; (**c**) Vp map from DCE study; (**f**) CBV map from DSC perfusion study. Right parietal lesion with irregular enhancement, cystic component and large NEPA. In the Vp map (**c**), ROI 1 positioned in the NEPA (arrowhead) shows a higher value compared to ROI 2 positioned in the contralateral normal-appearing white matter (ROI 1 Vp = 4.412 × 10^−3^; ROI 2 Vp = 0.276 × 10^−3^). The CBV map (**f**) shows an increase in the CBV value (white arrow) in the NEPA compared to the contrast-enhanced area, suggesting an infiltrative pattern. This case is also shown in Figure 3. PWI: perfusion-weighted imaging; HGG: high-grade glioma; Vp: volume of plasma; DCE: dynamic contrast-enhanced; CBV: cerebral blood volume; DSC: dynamic susceptibility contrast-enhanced; NEPA: non-enhancing peritumoral area; ROI: region of interest.

**Figure 9 cancers-15-02992-f009:**
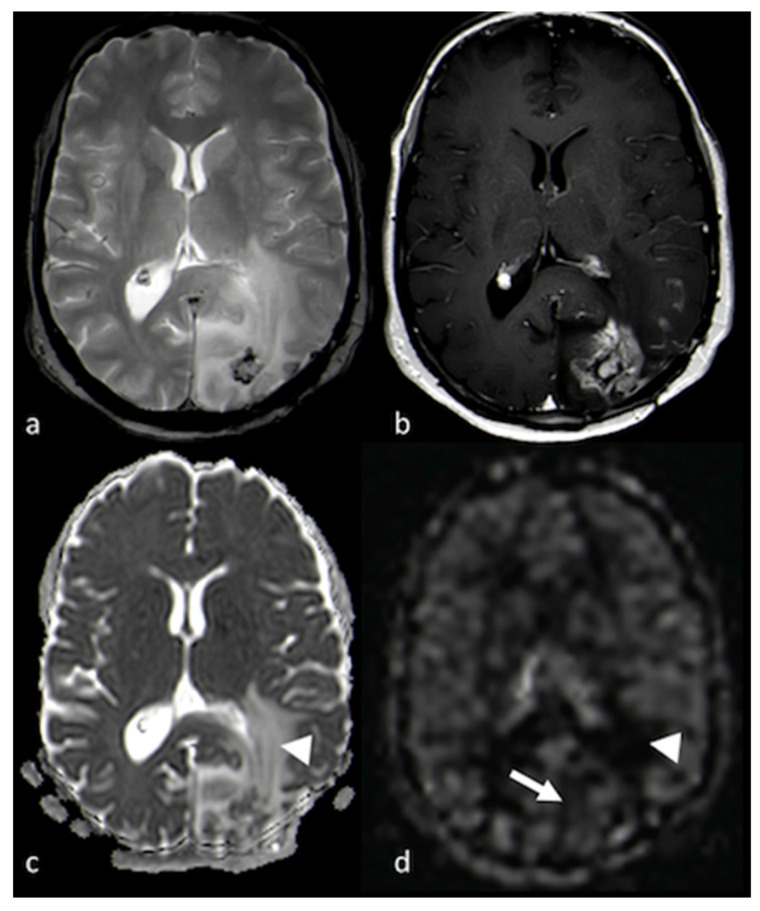
DWI and ASL in a left occipital BM: (**a**) axial T2 GE-weighted image; (**b**) axial postcontrast T1-weighted image; (**c**) ADC map from DWI study; and (**d**) CBF map from ASL study showing a left occipital BM with enhancement and large NEPA, with high CBF in the contrast-enhanced lesion (white arrows) and high ADC and low CBF within the NEPA (white arrowheads), suggesting vasogenic edema. DWI: diffusion-weighted imaging; ASL: arterial spin labeling; BM: brain metastasis; GE: gradient echo; ADC: apparent diffusion coefficient; CBF: cerebral blood flow; NEPA: non-enhancing peritumoral area.

**Figure 10 cancers-15-02992-f010:**
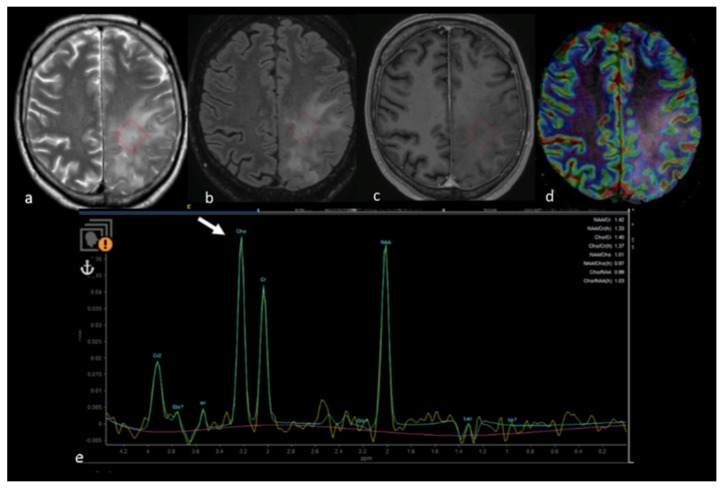
MRS study of left parietal HGG: (**a**) axial T2−weighted image; (**b**) axial T2 FLAIR image; (**c**) axial postcontrast T1−weighted image; (**d**) CBV map from DSC imaging study; and (**e**) MRS single−voxel study with 144 ms echo time showing an increase in the choline peak (white arrow) in the NEPA even in the absence of a CBV increase, suggesting an infiltrative pattern in the NEPA. The red box is the VOI used for MRS study. MRS: magnetic resonance spectroscopy; HGG: high-grade glioma; CBV: cerebral blood volume; DSC: dynamic susceptibility contrast-enhanced; NEPA: non-enhancing peritumoral area; VOI: Volume of Interest.

**Figure 11 cancers-15-02992-f011:**
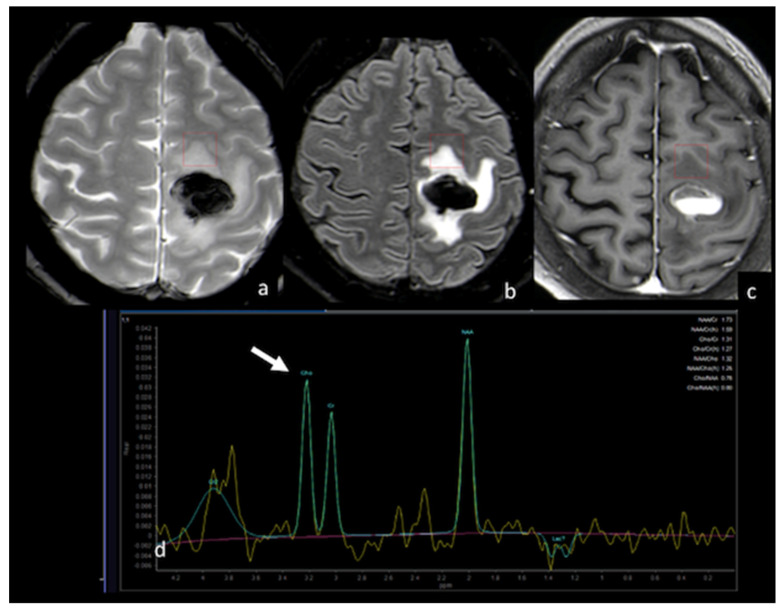
MRS study of left frontal hemorrhagic BM: (**a**) axial T2 GE−weighted image; (**b**) axial T2 FLAIR image; (**c**) axial postcontrast T1−weighted image; (**d**) MRS single−voxel study with 144 ms echo time showing a normal choline peak (white arrow) in the NEPA, suggesting a vasogenic pattern in the NEPA. The red box is the VOI used for MRS study. This case is shown also in Figure 1. MRS: magnetic resonance spectroscopy; BM: brain metastasis; GE: gradient echo; NEPA: non-enhancing peritumoral area; VOI: Volume Of Interest.

**Figure 12 cancers-15-02992-f012:**
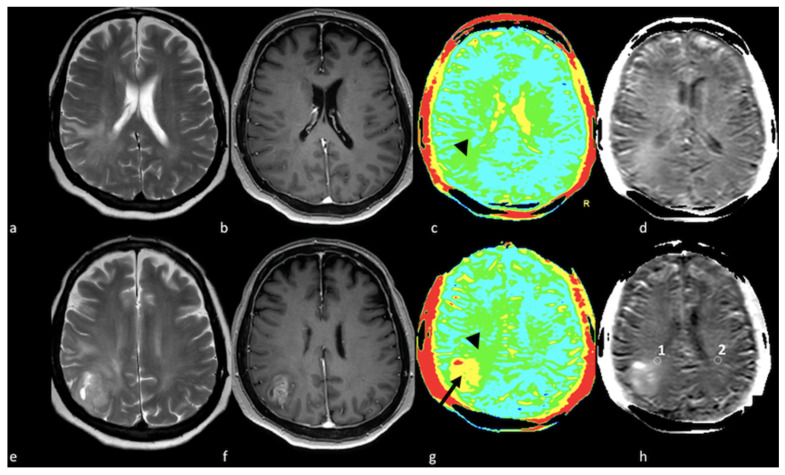
APT in right parietal HGGs: (**a**,**e**) axial T2-weighted image; (**b**,**f**) axial postcontrast T1-weighted image; (**c**,**g**) color APT map; (**d**,**h**) APT map. Right parietal lesion with irregular enhancement, cystic component and associated NEPA. In the APT maps (**c**,**d**,**g**,**h**), the enhanced area shows increased APT (arrow); (**c**,**d**) increased APT values were also demonstrated in the NEPA (arrowhead), suggesting an infiltrative pattern, as confirmed by increased APT values measured in ROI 1 in NEPA (1.71%) compared to ROI 2 (0.65%) in left normal appearing white matter (**h**, white circles). This case is also shown in Figure 3. APT: amide proton transfer; HGG: high-grade glioma; NEPA: non-enhancing peritumoral area; ROI: Region Of Interest.

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
