# Peer review of "Conventional and Advanced Magnetic Resonance Imaging Assessment of Non-Enhancing Peritumoral Area in Brain Tumor"

_cancers, 2023, doi:10.3390/cancers15112992_

Round 1

Reviewer 1 Report

General comment:  In general this is a very thorough review of neuroimaging techniques for assessing the NEPA in brain tumors.  The review generally provides equal weight to each technique mentioned with some exceptions noted below. 

Comment 1: For each section, it would be helpful to provide a brief description of how the images are acquired to generate the image contrast.  For DTI vs DWI, you could discuss how a series of images are collected with gradients applied in different directions to enable computation of the diffusion tensor. 

Comment 2:  A general comment for all of the figures that depict repeated image types (e.g., Figure 3,4, 6,7).  Do these images represent just different slices within the brain or different visits?  Please either expand upon that in the caption or within the figure itself. 

Comment 3: Figure 3 and 4.    Can you arrange the image modalities in a similar order from figure to figure?  For example in Figure 3 you have ADC-Flair-Post Contrast while in Figure 4 it is Arranged as Post-contrast - ADC - FLAIR and the DTI image.  

Comment 4:  Section 3.3 Perfusion Imaging techniques--In this three paragraph section try to keep the balance between the amount of time/details you present on the three perfusion techniques equal.  There is very limited details on ASL and what type of information it can provide. I think expanding upon this paragraph and talk about the source of the signal/measurement in ASL.

Comment 5:   Lines 237 and 311.  Please be consistent on how you report Ktrans. (Ktrans vs K-trans) throughout the manuscript.   

Comment 6:  An example ASL image would be helpful in one of the perfusion figures. 

Comment 7: The distribution of figures per section is a little uneven.   For example there are no figures for section 3.6 (although you could refer to previous figures that show multi-parameteric imaging)  or section 4.

Comment 8: A potential approach to include is the use of mathematical modeling and multi-parametric MRI data to distinguish radio necrosis from progression in brain mets.  This effort from Weis et al observed that mechanical properties within the NEPA were important in separating radiation necrosis from tumor progression (https://doi.org/10.1002/mp.14999)

Comment 9:  Any additional section, or discussion in section 5 would be any newly developed techniques that have not yet reached widespread use but could show promise in the future .

Minor english editing may be needed, some typos  (line 151 sensibility instead of sensitivity, 155, strengthen vs strength) but the document is still very readable. 

Reviewer 2 Report

1. Fig. 9 should be presented in a more scientific style. Now it looks good for a scientific-popular magazine but not for a research journal.

2. It is suggested that further discussions and attributions regarding the proposed method and the findings of the experiments can be appended. Also, the current discussions and attributions should be logically reorganized and refined.

3. It is suggested that the conclusion section can also summarize all the findings and attributions, other than the introductions to the proposed methods.

I suggest the authors read studies performed by scholars such as Ranjbarzadeh et al., Jafarzadeh et al., and their groups

Reviewer 3 Report

In the present study, the authors have conducted an interesting review on the analysis of the peritumoral region in brain tumors using MRI. The paper is well-written, and the provided information is well-structured, clear, and easy to read. The topic is clinically relevant due to the relationship between the detected characteristics of the peritumoral microenvironment through MRI and patient outcomes. Although it would be ideal to present a systematic review on the subject, I suggest that the authors, at the very least, clarify that their work is a narrative review, as methodologically, they are distinct approaches.

Furthermore, I strongly suggest that the authors expand Section 3.6: "Multiparametric Approaches and Diagnostic Algorithms" and discuss the combination of artificial intelligence techniques and MRI for the detection of tumor infiltration and prediction of recurrent areas in glioblastoma. Given their relevance, the authors should include the following citations:

- Rathore, S.; Akbari, H.; Doshi, J. Radiomic signature of infiltration in peritumoral edema predicts subsequent recurrence in glioblastoma: Implications for personalized radiotherapy planning. J. Med. Imaging 2018, 5, 1.

- Akbari, H.; Macyszyn, L.; Da, X.; Bilello, M.; Wolf, R.L.; Martinez-Lage, M.; Biros, G.; Alonso-Basanta, M.; O’Rourke, D.M.; Davatzikos, C. Imaging surrogates of infiltration obtained via multiparametric imaging pattern analysis predict subsequent location of recurrence of glioblastoma. Neurosurgery 2016, 78, 572–580. 

- Yan, J.L.; Li, C.; van der Hoorn, A.; Boonzaier, N.R.; Matys, T.; Price, S.J. A Neural Network Approach to Identify the Peritumoral Invasive Areas in Glioblastoma Patients by Using MR Radiomics. Sci. Rep. 2020, 10, 1–10.

- Cepeda S, Luppino LT, Pérez-Núñez A, Solheim O, García-García S, Velasco-Casares M, Karlberg A, Eikenes L, Sarabia R, Arrese I, Zamora T, Gonzalez P, Jiménez-Roldán L, Kuttner S. Predicting Regions of Local Recurrence in Glioblastomas Using Voxel-Based Radiomic Features of Multiparametric Postoperative MRI. Cancers. 2023; 15(6):1894

- Juan-Albarracín J, Fuster-Garcia E, Pérez-Girbés A, et al. Glioblastoma: Vascular Habitats Detected at Preoperative Dynamic Susceptibility-weighted Contrast-enhanced Perfusion MR Imaging Predict Survival. Radiology. 2018;287(3):944-954.

- Riahi Samani Z, Parker D, Akbari H, et al. Artificial intelligence-based locoregional markers of brain peritumoral microenvironment. Sci Rep. 2023;13(1):963.

- Chougule, T.; Gupta, R.K.; Saini, J.; Agrawal, S.; Gupta, M.; Vakharia, N.; Singh, A.; Patir, R.; Vaishya, S.; Ingalhalikar, M. Radiomics signature for temporal evolution and recurrence patterns of glioblastoma using multimodal magnetic resonance imaging. NMR Biomed. 2022, 35, e4647.

- Dasgupta, A.; Geraghty, B.; Maralani, P.J.; Malik, N.; Sandhu, M.; Detsky, J.; Tseng, C.-L.; Soliman, H.; Myrehaug, S.; Husain, Z.; et al. Quantitative mapping of individual voxels in the peritumoral region of IDH-wildtype glioblastoma to distinguish between tumor infiltration and edema. J. Neurooncol. 2021, 153, 251–261.

Round 2

Reviewer 3 Report

The authors have diligently incorporated the suggestions made by the reviewers, significantly enhancing the overall quality of the manuscript.